# An Explainable Student Fatigue Monitoring Module with Joint Facial Representation

**DOI:** 10.3390/s23073602

**Published:** 2023-03-30

**Authors:** Xiaomian Li, Jiaqin Lin, Zhiqiang Tian, Yuping Lin

**Affiliations:** 1School of Foreign Studies, Xi’an Jiaotong University, Xi’an 710049, China; mianmianli@126.com; 2Institute of Artificial Intelligence and Robotics, School of Electronic and Information Engineering, Xi’an Jiaotong University, Xi’an 710049, China; ljq0306@stu.xjtu.edu.cn; 3School of Software Engineering, Xi’an Jiaotong University, Xi’an 710049, China; zhiqiangtian@xjtu.edu.cn

**Keywords:** online fatigue detection, video-based online fatigue detection, CNN, MPA, joint facial representation

## Abstract

Online fatigue estimation is, inevitably, in demand as fatigue can impair the health of college students and lower the quality of higher education. Therefore, it is essential to monitor college students’ fatigue to diminish its adverse effects on the health and academic performance of college students. However, former studies on student fatigue monitoring are mainly survey-based with offline analysis, instead of using constant fatigue monitoring. Hence, we proposed an explainable student fatigue estimation model based on joint facial representation. This model includes two modules: a spacial–temporal symptom classification module and a data-experience joint status inferring module. The first module tracks a student’s face and generates spatial–temporal features using a deep convolutional neural network (CNN) for the relevant drivers of abnormal symptom classification; the second module infers a student’s status with symptom classification results with maximum a posteriori (MAP) under the data-experience joint constraints. The model was trained on the benchmark NTHU Driver Drowsiness Detection (NTHU-DDD) dataset and tested on an Online Student Fatigue Monitoring (OSFM) dataset. Our method outperformed the other methods with an accuracy rate of 94.47% under the same training–testing setting. The results were significant for real-time monitoring of students’ fatigue states during online classes and could also provide practical strategies for in-person education.

## 1. Introduction

It has been reported that long-time online classes under the education policy of the COVID-19 pandemic can cause middle-level fatigue for college students [1]. Research indicates that higher fatigue levels can result in social isolation to some degree [2]. Both the state of online fatigue and the adverse impacts brought on by online fatigue can lower college students’ performance and grades, and the situation can be even more severe for students with poor physical or psychological conditions. Thereby, online fatigue estimation has become a timely tool because its estimation results can provide guidance for the teaching process and education administration.

At the beginning of the twentieth century, mass-produced technology began to take the place of primitive workshops in all walks of life [3]. The aviation and car industry boomed during this period and the fatigue of metal structures in airplane and car equipment became pivotal as airplanes and cars were promoters for mass production [4]. Since the middle of the twentieth century, a large quantity of fatigue research spontaneously targeted the detection of defects and the endurance of metal fatigue, or other material fatigue, to guarantee safe and top-speed industrial production [5]. Simultaneously, fatigue estimation in human operators became the focus of fatigue research as human resources were also vital factors for the high-speed rotation of industry [6]. Especially, the prosperity of the airplane- and car-driving business has made piloting fatigue one important area in fatigue research [7,8].

Nowadays, the tough academic demands in higher education puts college students under much more physical and mental pressure than ever before [9]. As a result, most college students are in a state of fatigue during class time, especially in the first class in the morning [10]. To date, questionnaires and surveys are the most frequently used methods in student fatigue research [11,12,13,14], but video-based deep learning is seldom used in this field. Based on previous piloting fatigue experiments, in this research, we devised a joint facial fatigue representation classification module and an explainable fatigue estimation module for online fatigue detection in college students. Firstly, our model uses video data for online student fatigue monitoring. Student states were captured directly from video data, thus, our monitoring results were more accurate and timely. Moreover, as the features from different fatigue-related regions in our model are fused with the fatigue state inferring process, the model is more robust and can distinguish and explain detailed fatigue degrees.

The rest of the paper is organized as follows. Section 2 briefly reviews the research concerning fatigue estimation. Section 3 elucidates the present explainable fatigue monitoring module. Section 4 discusses the advantages of the present student fatigue detection module compared to previous methods. In Section 5, the conclusion is presented.

## 2. Related Works

As fatigue estimation plays an important role in the biological recovery of the human body, enormous investigations, experiments, and research have been carried out to estimate fatigue by various approaches. To sum this up, these estimation approaches can be categorized into four types: survey methods, instrument testing methods, observational methods, and blended methods.

### 2.1. Survey Methods

Questionnaire and survey methods are the traditional methods that have been frequently applied in student fatigue research. Here are some examples.

Through questionnaires, researchers have testified that near work can cause visual fatigue, initiate myopia, and enhance the degree of myopia in college students [15,16,17,18,19,20,21,22,23]. Since the breakout of the COVID-19 pandemic, researchers have revealed that long-term online education can aggravate the situation of eye fatigue among college students [24,25,26,27]. Meanwhile, an online survey also found that viral infections and psychological stress can trigger chronic fatigue syndrome among college students [28].

### 2.2. Instrument Testing Methods

With the emergence of new materials and technologies, more and more innovative measuring instruments have been adopted in scientific, technical, and physiological examinations. At present, electroencephalograms (EEGs), electrocardiograms (ECGs), and electro-oculograms (EOGs) are widely used devices in testing physical and mental fatigue. These instruments are noninvasive and safe, and the data acquisition process is nearly in real-time. Moreover, the convenience and low cost of new wearable materials have greatly boosted the extensive usage of these instruments in detecting the fatigue state of individuals in different conditions and environments.

### 2.3. Observational Methods

By conscious noticing and detailed examination of participants’ behaviors in a naturalistic background, observation has been identified as a fundamental base of all research methods [29]. Observation is not only one of the most important research methods in social sciences but it is also one of the most complex ones [30]. It may be the main method in the project or one of several complementary qualitative methods for other types of research [31]. As fatigue detection and estimation belong to the research field of human behavioral science, many researchers have applied this method for collecting elements that influence the physical or mental states of drivers, students, workers, patients, or other populations. For example, Claudia Kedor and her partners observed 42 chronic fatigue syndromes in patients following the COVID-19 pandemic [32].

### 2.4. Blended Methods

A lot of research combines more than one method to study the fatigue state of different populations. In effect, many researchers combine an instrument testing method and an observational method together to study piloting fatigue, muscle fatigue, vision fatigue, and other types of fatigue. Nowadays, new techniques such as video technology and machine learning algorithms are combined to measure the scales and degrees of a fatigue state. Li and her peers stated that deep learning can easily collect complicated face features and dig out the latent rules of face images in certain settings [33]. As human face recognition can be more efficient and precise by adding an attentional mechanism [34], introducing attention to observation-based methods can also improve the generalization and robustness of the method. Machine learning/deep learning has been widely used to analyze visual fatigue among college students [35].

In this research, we used an observational method by collecting long simulated raw videos of college students presenting with normal and fatigue behavior, as observational methods can provide real-time monitoring and, thus, make the research more reliable. These videos were then divided into 2474 clip samples that were processed by machine learning in order to study the online fatigue of university students compared with piloting fatigue.

## 3. Our Proposed Approach

### 3.1. The Proposed Framework

Facial fatigue symptoms include head nodding (head anomaly), yawning (mouth anomaly), and blink frequency (eye anomaly) [36]. However, due to the diversity of equipment in the online detection scene, factors such as viewing angle and the line of sight are likely to interfere with the detection of head movements. Moreover, head movements are difficult to detect and this can lead to low reliability. Therefore, the two major fatigue symptoms in online classes are eye anomalies and mouth anomalies, which are also highly rotation-irrelevant. Inferring a student fatigue state simply with data to fatigue mapping is neither reliable nor explainable. It is hard to determine different fatigue degrees in the monitored subject and act based on these monitoring results. Therefore, we proposed the monitoring of a student fatigue state using facial fatigue representations, such as eye and mouth anomalies, then interpretably fusing all the fatigue symptoms with the data-driven maximum posterior rule and human experience constraints.

Our model monitors the two facial representation subsets: the eye state representation and the mouth state representation. In each representation set, we distinguish the anomaly representation from the normal representation with a spatial–temporal feature extraction network constructed with temporal shift module (TSM) [37] blocks. The extracted fatigue representations of both subsets are later fused with a naive Bayesian model to predict the overall fatigue state.

Our network comprises a facial fatigue representation classification module and an interpretable fatigue estimation module. As shown in Figure 1, in the facial fatigue representation classification module, the network takes preprocessed frame sequences as input and generates the facial fatigue representation on both subsets.

### 3.2. Fatigue Representation Classification

Prior to sharing human research data, the eye and mouth region actions can most significantly represent the human fatigue state. Therefore, we proposed a facial fatigue representation classification network built with TSM blocks to extract the fatigue-relevant features in these two key areas.

In this module, the proposed network first performed facial detection on the raw video input to enhance the task-relevant region and suppress background noise. This procedure has also been proved to be beneficial to the robustness of the proposed model.

Considering that the target fatigue symptoms in the eye or mouth regions are also microactions, we utilized TSM blocks as the basic feature extraction unit to take advantage of state-of-the-art video action recognition approaches. In these units, the modeling of temporal information in the input sequence is obtained by a short-range feature shift in the temporal dimension. In the facial fatigue representation learning module, three TSM blocks, based on ResNet50 [38], are utilized as the spatial–temporal representation extractor.

Moreover, to enhance the generalizability and robustness of our model, we trained our model on a driver fatigue detection benchmark dataset, which contained a larger amount of data and more scenario variants. We adopted this training–testing procedure for the following three reasons. Firstly, fatigue research belongs to personnel fatigue, and drivers and students are consistent in their behavioral representation. Secondly, the research method extracted the facial data and this avoided the influence of environmental background and other factors, which can enhance the robustness of the algorithms on different tasks. Lastly, the low volume of relevant datasets for student fatigue detection can be complemented by the driver fatigue dataset.

To validate the generalizability of the trained model, we tested our model on an independent dataset without fine-tuning. In the training process, the sigmoid-crossentropy loss was applied for the multi-label facial representation classification of extracted face sequences. Since sigmoid-crossentropy allows multiple positive classes, in the anomaly detection stage, it was used to minimize the distance between the multilabel representation ground truth and the prediction. Moreover, the contributions of these independent classes were weighted separately with class weight factors, $p_{j}$. The loss functions can be expressed as in Equation (Equation 1).
(1)$$\begin{array}{c} loss_{s}=\frac{1}{N}\sum_{j=1}^{C}\sum_{i=1}^{N}-\left[y_{i,j}p_{j}\log\sigma_{g}\left(x_{i,j}\right)+\left(1-y_{i,j}\right)\log\left(1-\sigma_{g}\left(x_{i,j}\right)\right)\right]+N\left(\lambda \right), \end{array}$$
where $N,C,y_{i,j},x_{i,j},p_{j},\lambda ,N\left(\lambda \right)$ are the batch size, class number, *j*-th class prediction and label of the *i*-th sample, *j*-th class weight, network parameters, and $L_{2}$-regularization coefficient, respectively. The value $\sigma_{g}$ is the activation function, Sigmoid. The values $y_{i},x_{i}$ are the label and prediction of the *i*-th sample in each batch.

### 3.3. Interpretable Fatigue Estimation

Different fatigue symptoms represent different degrees of fatigue. Detailed fatigue symptom detection results can provide comprehensive information about a student’s state of fatigue. In addition, purely data-driven methods such as support vector machines (SVMs) are susceptible to noisy data, while purely experience-driven methods such as thresholds have poor adaptability towards new environments. Compared with other complex classification algorithms based on a large amount of training data, the naive Bayesian classification algorithm has a better learning efficiency and classification effect, and it is widely used in text classification, content classification [39,40], and so on. Therefore, considering the particularity of state data labeling and the requirements for continuous detection and detection efficiency in student fatigue detection application scenarios, we utilized a MAP model constrained by experience rules to estimate the student fatigue state based on detected symptoms.

In the MAP model, the posterior probability is linked to the appearance frequency, for which the noise samples tend to be suppressed by probability averaging. Therefore, the MAP model is more robust under variate backgrounds or environments. To simplify the status inferring problem, we assumed that the symptoms of different subsets were approximately independent. Thus, we could estimate the drowsiness state with two approximately independent attributes under the maximum posterior rule. For a fatigued state, *s*, the observable symptoms set, $A=\{a_{e},a_{m}\}$, and the parts subsets, P={e,m}, the MAP problem could be constructed as in Equation (Equation 2):(2)$$\begin{array}{cc} \max_{s}\; & P\left(s|a_{e},a_{m}\right)=\frac{P\left(s\right)\prod_{i\in P}P\left(a_{i}|s\right)}{P(a_{e},a_{m})}.\\ s.t.\;\; & a_{m}a_{e}\left(s-1\right)=0. \end{array}$$
(3)$$\begin{array}{cc} P(a_{i}|s)= & \frac{1}{\sqrt{2\pi \sigma_{s}^{2}}}\exp\left(-\frac{{\left(a_{i}-\mu_{s}\right)}^{2}}{2\sigma_{s}^{2}}\right),i\in P. \end{array}$$
where $P(s|a_{e},a_{m})$, $P(a_{e},a_{m})$, and $P(a_{i}|s)$ are the posterior probability, the prior probability of symptoms, and symptoms under each state, respectively. The prior Gaussian distribution is determined by the mean $\mu_{s}$ and variance $\sigma_{s}$. The prior probability of attributes is approximated with a Gaussian distribution as in Equation (Equation 3). The MAP model was trained with the symptom labels of the training dataset. Additionally, as shown in (Equation 2), certain experience constraints constructed by human experience were applied to the MAP model to constrain the fatigue state result with eye and mouth condition sequences.

## 4. Experiments

### 4.1. Dataset and Implementation Protocols

#### 4.1.1. NTHU-DDD Dataset

We trained our model on the driver fatigue detection benchmark dataset, the NTHU Driver Drowsiness Detection (NTHU-DDD) dataset [41], which contains videos of 36 sampled subjects of different races in a simulated driving environment, a normal driving environment, yawning, and exhibiting sleepiness in various scenes with different facial decorations. The training set of the NTHU-DDD dataset contains 18 individuals that were separated from the testing set. The original videos were preprocessed and divided into clips before training. The frame samples of the five scenarios are shown in Figure 2.

#### 4.1.2. OSFM Dataset

We prodcued a dataset of videos, named the Online Student Fatigue Monitoring dataset, for model performance evaluation. The videos were collected with different mobile devices to simulate real scenarios. The original data included videos recorded by both male and female students at different times and angles. These video data contained simulated normal behavior and fatigue behavior, including yawning and sleepy eyes. To enrich the dataset, some samples were taken from the same subject in different scenarios and times. As a result, we collected 41 long raw videos from 34 subjects, which were later divided into 2474 clip samples for video-wise prediction and evaluation. The frame samples of different behaviors of different subjects are shown in Figure 3.

#### 4.1.3. Implementation

To take advantage of the rich variance of the data variance, such as scenario, illumination, and gesture, we trained our model on the benchmark NTHU-DDD dataset and tested it on the OSFM dataset without fine-tuning. The sample distribution is shown in Table 1. We implemented our model with Pytorch on two Nvidia 3090 GPUs for both training and evaluation. The feature backbone of our model was fine-tuned from the ImageNet [42] pre-trained weights. At the clip level, we sampled input frames using temporal sampling network (TSN) sampling [43] following the TSM. The sample clips from raw videos were sampled by uniform sampling. We trained the model using a stochastic gradient descent (SGD) optimizer with a learning rate of $2.5\times 10^{-3}$ for 120 epochs.

### 4.2. Comparison of Fatigue State Estimation Techniques

In this section, the students’ fatigue state inference results are shown in Table 2 and we compared the results with other fatigue detection methods using the OSFM dataset. The face sampling network (FSN) [44] samples key areas from the face for feature fusion and state prediction; the ensemble network (E-Net) [45] collects different features extracted by a group of backbones to generate state predictions. We provided frame-wise precision, recall, and an *f*1-score of the drowsy class, along with the accuracy of overall samples in all test models for a fair comparison. As shown in Table 2, this model showed both higher precision and more excellent recall, which testifies that this new method is much more accurate and sensitive than previous fatigue detection methods. The metrics were calculated as in Equation (Equation 4) and we bolded the optimal results of the same subset.
(4)$$\begin{array}{cc} precision= & \frac{TP}{TP+FP}\\ recall= & \frac{TP}{TP+FN}\\ f1-score= & \frac{2\cdot precision}{precision+recall}\\ accuracy= & \frac{TP+TN}{N} \end{array}$$
where TP,TN,FP,FN,N are the true positives, true negetives, false positives, false negatives, and the total number of all samples.

To further testify the advantage of our design, we also evaluated the model performance on the subset level. The multilabel classification accuracy on the subset level is shown in Table 3, which manifests that on both subsets our model can classify anomalous behaviors more accurately and sensitively. The receiver operating characteristic curve (ROC curve) of our facial representation classifier is exhibited in Figure 4. For both subsets, our model presented high true positive rates on different false positive rates.

As mentioned above, our model was trained on the NTHU-DDD dataset and tested on the OSFM dataset without fine-tuning. The experimental results supported our claim of the high robustness of the proposed model. The robustness of the proposed model was gained from both the rich variance of the training dataset and the structural similarity of the human face. We summarized the subset detection accuracy of all subjects in the OSFM dataset. As exhibited in Figure 5, where the darker color denotes higher accuracy, the model’s performance varied over the subjects but held a high detection accuracy towards most subjects, which indicated that our method achieved an excellent overall performance. We visualized the gradient class activation heat map (Grad-CAM) [46] of our backbone model for samples of eye anomalies, mouth anomalies, and eye-head anomalies. As exhibited in Figure 6, the feature backbone was highly sensitive to mouth region anomalies and could effectively deal with scenarios with facial decorations, such as glasses.

To guarantee the explainability and the advantage of our design, we further visualized the subset prediction probability–time curve, which is presented in Figure 7. The horizontal axis represents the time dimension, and the vertical axis represents the predicted confidence. As exhibited in the curve, as the subject performed a series of anomalous behaviors in each subset, the model captured a group of probability peaks that could be regarded as the center of the anomalous behaviors. Additionally, as our method uses both spatial and temporal information, our video-based results were more stable and smoother than frame-based predictions.

To sum up, in this research, we established an effective explainable student fatigue monitoring module with joint facial representation. Through analyzing the collected videos by machine learning, our research found that our module can better recognize the online fatigue state of individual students and can also differentiate their fatigue degree. Our method achieved a higher accuracy rate of 94.47% than the other methods under the same training–testing setting.

## 5. Discussion

With the renewal of technology, fatigue research has evolved from a qualitative perspective to a quantitative perspective. The major approaches applied in the previous physical and mental fatigue research mainly include surveys, instrument testing, observational, and blended methods. Among these methods, surveys play a major role as they can be repeatedly used by researchers. In fatigue research, many researchers select surveys, especially questionnaires, as their research method due to this convenience [47,48,49,50,51]. Take De’s research as an example, the author investigated 541 medical students and their investigation found that the mean prevalence of overall online fatigue was 48%. Similar to De’s survey, other surveys also demonstrated that online fatigue is a universal phenomenon among college students, but none of them described the students’ fatigue states in detail.

In surveys, researchers usually use statistical software such as SPSS to analyze the data, but this data management can just give a numerical presentation on the fatigue prevalence among individuals or the differences in occurrences between different populations. In this way, the potential factors that influence the subject may be picked out, but the data and the research results rely much more on the participants’ self-evaluation. Therefore, it is not completely possible to guarantee the research validity. Additionally, the late workload of data management can sometimes be overwhelming in time and effort for its analysis.

As mentioned before, various electrical instruments such as EEGs, ECGs, and EOGs have been widely used in detecting physiological fatigue. For example, in the work of Marcin and his peers, they used EOGs to detect students’ fatigue states by the index of eye anomalies, but their research just used eye symptoms to illustrate the condition [32]. Moreover, during the experiment, the monitoring participants must be wearing those instruments on their bodies and an attached display shows the brain waves, heartbeats, and eye-moving curves. These instruments are usually expensive and, thus, it is difficult to involve a large number of samples. At the same time, the participants must keep certain postures, which can be quite uncomfortable as body positions are especially hard to control in fatigue states. Despite these disadvantages, these electrical instruments can accurately signal a fatigued state, even minute changes in the fatigue state.

Compared with the prior methods, the video-based observational method we have used in this research has a lot of advantages. First and foremost, the monitoring can be real-time and it is harmless to the participants. It is easy to control and operate compared to electrical instruments. Secondly, what matters most is that nearly no interference disturbs the research procedures and results. Additionally, due to the popularization of high-definition monitoring cameras in university classrooms, genuine samples and data that are required for our research can be collected more easily than data collected in surveys.

Another advantage of our fatigue estimation method is the usage of artificial intelligence and machine learning. Artificial intelligence has been widely used to detect fatigue among drivers [52,53,54,55,56], but it has seldom been used to detect student fatigue. Nevertheless, online education prevails around the world due to the COVID-19 pandemic. Under this global situation, several pieces of research have revealed that online class environments are much more likely to cause fatigue than traditional classroom environments [57,58,59]. After all, college students are now attending classes in places such as bedrooms or dining rooms, which obviously lack the aura of studying in classrooms. Lack of monitoring from teachers and class engagement of activities may further boost the sense of sleepiness.

In a conference in 2010, Rabey Husini et al. reported that their research took the blink rate and duration as measures of fatigue and mental workload [60], but this research only depended on one symptom, eye anomalies, to estimate students’ fatigue. In the research of Diah and his peers, they used EEG technology to monitor 13 senior high school students to recognize the mental fatigue condition. Obviously, the research numbers were very low and the analysis method used was Pearson correlation, which only confirmed the presence of fatigue conditions after the workload of a full day of school.

However, in our research, we used monitored facial presentations to study the online fatigue of college students. To be specific, we monitored the students’ fatigue states based on facial fatigue representations, such as eye anomalies and mouth anomalies. We then interpretably fused all the fatigue symptoms with the data-driven maximum posterior rule and human experience constraints. The fusing method could better decipher a student’s fatigue state in a comprehensive way, which is the novelness of our method. In addition, both the multiple representations and fusion method made our video-based results more stable and smoother than frame-based predictions, as both spatial and temporal information was processed.

## 6. Conclusions

For general fatigue, there is a disability for performing either mental or physical work, and this disability is noticed first in work requiring attention and sustained effort and, lastly, in those acts that have become automatic [61]. Online fatigue, one type of general fatigue, has a great influence on college students’ health and performance, as the academic load of college students is rather large with the speedy renewal of knowledge and technology. For these reasons, fatigue estimation is a very important aspect of fatigue research because the research results can provide concrete guidance to design proper class hours, as well as proper study content for students at different ages. For example, in 1899 Kratz investigated how to reduce fatigue to the minimum in the schoolroom and the results showed that it might be quite safe to require college students to devote at least as much time as is ordinarily taken for examination periods without the fear of either doing them an injustice from the point of view of measurement or of tiring them unduly [62].

Fatigue state estimation has broad applications and business potential in autopiloting, industrial worker monitoring, education, etc. In this research, we established an effective explainable student fatigue monitoring module with joint facial representation. By simulating fatigue models and machine learning, our research not only recognized the online fatigue state of individual students but it also differentiated their fatigue degrees. These results are very beneficial for higher education management, as well as online education, in order to be able to design more specific teaching plans for students with different grades, majors, or sexes. Therefore, our further study will focus on fatigue differentiation technologies among college students in different majors. 

## Figures and Tables

**Figure 1 sensors-23-03602-f001:**
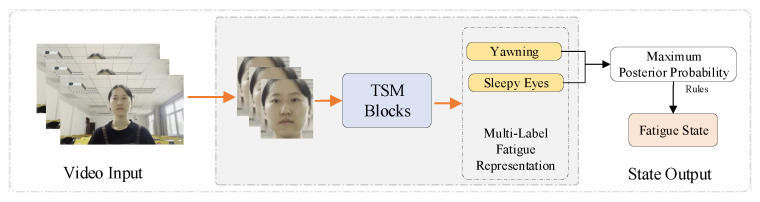
Architecture of the interpretable student fatigue monitoring network. The proposed network includes an anomaly classification module and a Bayesian fatigue state inference module.

**Figure 2 sensors-23-03602-f002:**
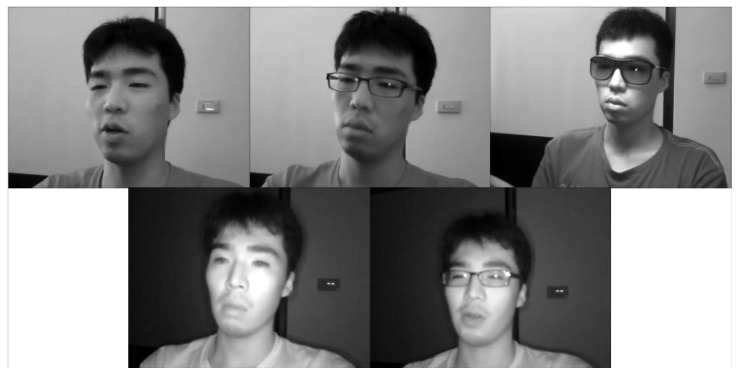
Frame samples of five scenarios in the NTHU-DDD dataset. The images in the first row are sample frames taken in daylight, and from left to right are the no-glasses, glasses, and sunglasses scenarios, respectively. The images in the second row are sample frames taken at nighttime.

**Figure 3 sensors-23-03602-f003:**
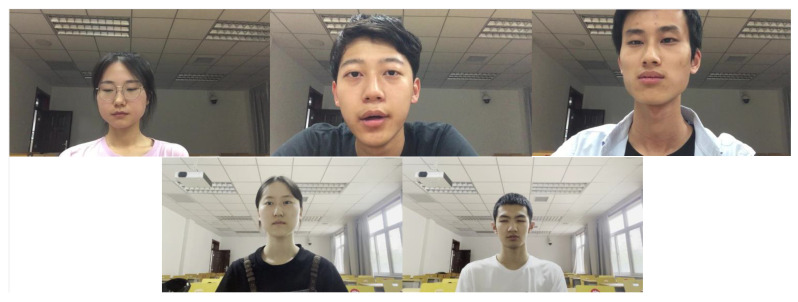
Frame samples of different subjects in different scenarios and at different angles. The first row are samples with darker backgrounds and a closer view; the second row are samples with better-illuminated backgrounds and a distant view.

**Figure 4 sensors-23-03602-f004:**
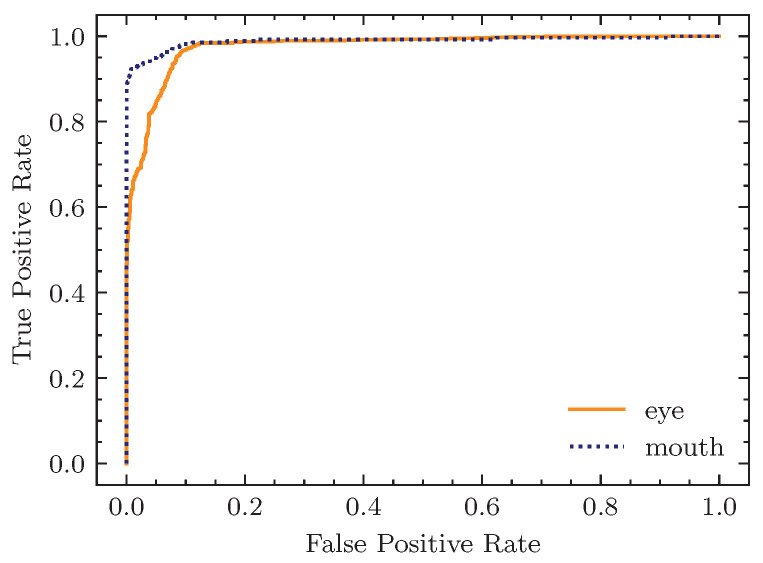
The ROC curve of our facial representation classifier.

**Figure 5 sensors-23-03602-f005:**
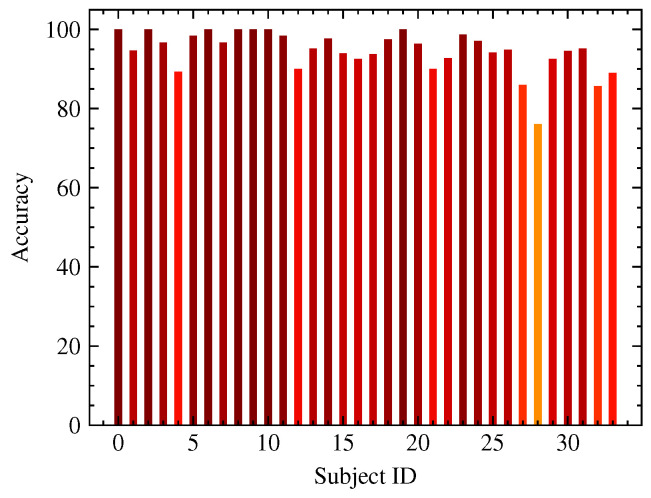
The subset detection accuracy of all subjects in the OSFM dataset. The horizontal axis is the subject ID in the OSFM dataset and the vertical axis is the frame-wise fatigue state accuracy of the subjects.

**Figure 6 sensors-23-03602-f006:**
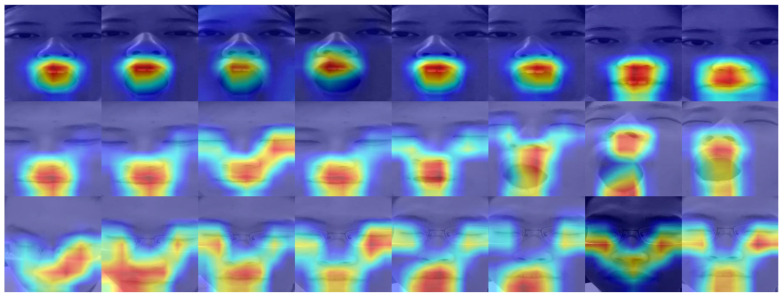
The class activation heatmap of our backbone model on samples of eye anomalies, mouth anomalies, and eye-head anomalies. The contribution of each region is represented by blue to red from low to high.

**Figure 7 sensors-23-03602-f007:**
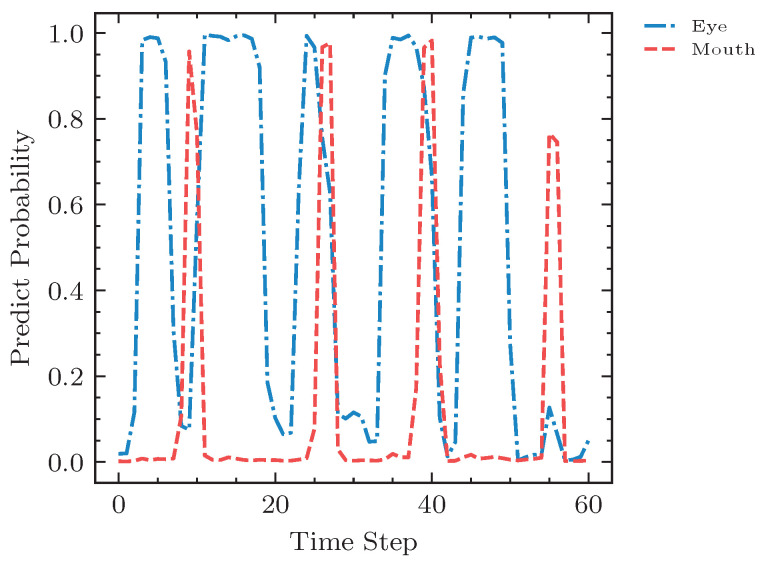
The factor analysis of the video sample from subject 005. The red and orange curves represent the predicted mouth and eye anomaly probabilities, respectively. The time–serial curves show the prediction results over time for a series of fatigue behaviors on the test video sample.

**Table 1 sensors-23-03602-t001:** Train and test sample distributions.

Dataset	Train	Val	Test
NTHU-DDD	44,138	11,321	-
OSFM	-	-	2474

**Table 2 sensors-23-03602-t002:** Performance comparison.

Model	Precision	Recall	*f*1-*Score*	Accuracy
E-Net	68.21	88.73	77.13	71.21
FSN	84.65	72.61	78.17	81.74
**Ours**	**93.90**	**97.29**	**95.60**	**94.47**

**Table 3 sensors-23-03602-t003:** Subset performance analysis.

Subset	Precision	Recall	*f*1-*Score*
eye	0.94	0.98	0.96
mouth	0.99	0.88	0.93

## Data Availability

The data used to support the findings of this study are available from the corresponding author upon request.

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
