# Peer review of "An Explainable Student Fatigue Monitoring Module with Joint Facial Representation"

_sensors, 2023, doi:10.3390/s23073602_

Round 1

Reviewer 1 Report

Explainable students fatigue monitoring with joint facial representation

Line 10: “The model is trained on a benchmark dataset and tested on a student fatigue dataset collected by us.” Interesting to see.

Line 125: “Therefore, 125

we propose to monitor the student fatigue state based on facial fatigue representations 126

like eye and mouth anomalies, then interpretably fuse all the fatigue symptoms with the 127

data-driven maximum posterior rule and human experience constraints.”

Why only these anomalies are used? Why not others?

Figure 1. is missing, author is advised to proofread the paper before submission,

Standard data set taken is NTHU, but used is for students activities and expression?

What is similarity between two, why it is used as a standard?

Figure 2, Figure 3 all are missing.

Such ignorance is not expected…..

Reject

Author Response

Dear reviewer, 

        Thank you a lot for your great comments. The response is in the attached file.

Sincerely, 

Xiaomian Li

Reviewer 2 Report

This is an interesting manuscript on Online fatigue estimation method based on face parameters.

1. The aim of the manuscript is the strongest part of the manuscript, original and practical. However, there are other points that, if improved, would provide a more impactful manuscript. My main concern is the theoretical framework. Many times paragraphs are described that are more historical way regarding to content than inherent  way to the topic under study. For example, the description of technological production at the beginning of the century corresponds more to a manual than to a scientific manuscript. Authors such as Hodge or Galton are described, which are not only from the beginnings of other sciences in other centuries, but also distance the update from the state of the art. It is imperative that authors update their sources to what is being worked on in this field, if possible in the current year or recently.

I would suggest some literature comparing human versus AI face recognition in this front. Some suggestions are described as follows:

Li, L., Mu, X., Li, S., & Peng, H. (2020). A review of face recognition technology. IEEE access8, 139110-139120.

Moret-Tatay, C., Fortea, I. B., & Sevilla, M. D. G. (2020). Challenges and insights for the visual system: Are face and word recognition two sides of the same coin?. Journal of Neurolinguistics56, 100941.

2. Why authors monitor the eye and the mouth state representation? More literature is needed in this front.

Same line 140-41 ", the eye and mouth region actions can most 140 significantly represent the human fatigue state". This state needs to be supported by literature.

3. Why a a Naive Bayesian model? More literature is needed to support this.

4. How it is possible that authors have 41 long raw videos from 34 subjects? This should be explained.

5. I don't understand why the authors name EEG when the objective of their study does not employ this technique. Moroever, the cite "Marcin 252 and his peers" is not adequate.

-There is no figure 1, 2 and 3

Author Response

(The authors gave the same response as above.)

Reviewer 3 Report

This manuscript proposes a video-based student fatigue estimation method to monitor the fatigue state during online classes. This study is timely and important for addressing the importance of fatigue research that is related to students’ health and academic performance when having online education. The core idea seems interesting, but the paper should be improved in some regards:

  1. The abstract should be improved. What is the motivation to study? Is that because there is no constant fatigue monitoring research? What will the gap be addressed? Should highlight what benchmark datasets are used in this paper. Please explain & justify the important methodology for the proposed model. Please give full name for CNN and MAP.

  1. Section 2. Line 67: ‘three types’ should be ‘four types’? Please briefly explain & justify using observational method in this research?

  1. Section 3. Line 136: Please elaborate in details of Figure 1. Line 161: Please give the full name of MAP. Line 164: Please check the probability equations. Could the author(s) show the proposed approach in a flowchart?

  1. Section 4. Preferably to use “Experiments & Results”. Line 193: Tag. 1 should be Tab. 1? Line 196: TSN? Line 198: SGD? Eq. 4: Where is accuracy? Table 3: Any comparison between models? Fig. 7: Hardly to see the difference colour & both dot lines look same. Results: Would like to see more implication and justification of the findings for all the analysis for several performance comparison / model rather than just reporting the results.

  1. Section 5: line 271: What is “57?” Please remove ‘?’. Line 278: Should be “et al.”. Line 285: We? Or we? Please briefly explain the findings to the proposed approach.

  1. References: Reference [44]: Tsm should be TSM. 

Author Response

(The authors gave the same response as above.)

Reviewer 4 Report

Dear Author, 

Your manuscript is well-written, and I have only a few minor suggestions for improvement. You need to fix those comments before your manuscript will be considered for publication.

Review comments to the author (SENSORS-2259849)  

In this manuscript, the author wrote an article entitled “Explainable students fatigue monitoring with joint facial representation” suitable for publication, but the concerned author has to rectify the below-mentioned minor review comments in the “Sensors”.

After rectifying the following reviewer’s minor comments, this article may accept. However, please rectify the following,

1.   What can we learn about a driver's fatigue level from observations of his or her face and head? What is the name of the system that detects drowsiness in real time in drivers by utilising facial features?

2.   What kind of artificial intelligence-based solution is there for detecting drowsiness and fatigue in drivers? Which algorithm is utilised when detecting the lane lines?

3.   Which facial feature is utilised by the drowsiness monitoring system that operates in real time?

4.   Which method of observation allows for the observation of the present behaviour of the respondents as it is taking place?

5.   Which of the primary methods of sampling allows the observer to decide in advance which types of behaviour they are interested in observing and then records every instance of those behaviours?

6.   What is the name of the method of sampling in which we select participants in such a way that each individual in the population has an equal chance of being selected, and the name of this method?

7.   What does the Bayesian inference model entail? What is an illustration of the Bayesian approach to inference?

8.   What exactly are these fatigue parameters? How many different approaches are there to fatigue analysis? 

9.   How do I locate the most effective model for deep learning? When evaluating a model, how do you determine how accurate it is?

10.   When and how does fatigue impair performance? How does exhaustion influence learning?

11.   What are the different ways that mental fatigue can be measured? Which tools are used to evaluate the effects of fatigue?

12.   What are the various kinds of electrodes that are utilised during an electrocardiogram, electroencephalogram, and electromyogram measurement?

It would be best if you rectify the above comments and submit them once again for your expectation. 

Author Response

(The authors gave the same response as above.)

Round 2

Reviewer 1 Report

ok changes are done significantly